# Belief-Aware Collaborative AI: Planning with Human Beliefs of AI Intentions

## Abstract

To enable effective human-AI collaboration, optimizing AI performance in isolation is not sufficient. AI systems need to also account for human factors. Prior research shows that incorporating models of human behavior into AI design can improve collaborative performance. However, existing approaches often implicitly assume that human behavior remains fixed regardless of the AI agent's actions. In practice, humans adapt their behavior based on their beliefs about the AI's intentions, that is, what they believe the AI is trying to accomplish. In this work, we develop and evaluate collaborative AI agents that account for human beliefs about AI intentions when choosing their actions. We formulate human-AI collaboration as a goal-oriented multi-agent decision-making problem and develop a belief model by extending level-$k$ reasoning with data-driven models of human behavior. Building on this belief model, we first design explicable AI policies that generate behavior from which humans can more easily infer the AI's intentions, providing a direct test of whether the model captures human belief formation. We then incorporate the belief model into the training of collaborative AI agents to improve coordination with human partners. Through simulations and extensive human-subject experiments, we show that our belief model better captures human inferences about AI intentions and can be used to generate more explicable AI behavior. More importantly, we demonstrate that collaborative AI agents trained with models of human beliefs significantly improve team performance in human-AI collaboration settings. These results demonstrate the value of modeling human beliefs about AI intentions as a design principle for collaborative AI.

## 1 Introduction

The potential for human-AI collaboration is immense and spans domains including healthcare (Cheng et al., 2016; Mobadersany et al., 2018), industrial manufacturing (Bauer et al., 2008; Sherwani et al., 2020), and workflow productivity (Wilson & Daugherty, 2018). In these settings, AI systems are increasingly expected not only to perform accurately but also to work alongside humans. However, despite significant improvements in AI performance over the past decade, designing AI agents that effectively collaborate with humans in a coordinated and reliable manner remains a challenge.

In particular, optimizing the AI system in isolation is not sufficient for enhancing the performance of human-AI collaboration. To optimize collaborative performance, AI agents need to account for their human counterparts when deciding on their strategies. For example, in AI-assisted decision-making, researchers have demonstrated that, instead of optimizing AI independently, training AI to focus on improving in areas where humans typically struggle can significantly enhance the performance of human-AI teams (Bansal et al., 2021; Wilder et al., 2021). In human-AI collaboration, Carroll et al. (2019) show that incorporating models of human behavior into the training of AI agents leads to higher collaborative performance compared to training the agent through self-play (Silver et al., 2018). These studies highlight the importance of integrating human behavior into the design of collaborative AI. However, a key limitation of these research efforts is that they mostly assume that human behavior remains static, irrespective of the actions and behavior of the AI counterpart. In practice, humans may modify their behavior in response to their beliefs about the intentions of AI agents based on their observations of AI behavior.

In this work, we argue and aim to demonstrate that, in addition to incorporating human behavior, designing AI agents that integrate human beliefs about AI intentions can further improve collaborative performance. Many existing approaches model humans as agents whose actions depend on observable environmental states or historical behavior. However, humans also form beliefs about what the AI intends to do and adjust their behavior accordingly, especially in settings without direct communication. These belief updates can substantially shape coordination dynamics. When humans adapt their actions based on inferred AI intentions, AI agents that explicitly account for such belief formation can achieve better collaborative performance than agents that model only human behavior. By reasoning about how their own actions shape human beliefs, AI systems can choose actions that facilitate more efficient coordination.

Our central research question is whether collaborative AI agents trained with models of human beliefs can achieve higher team performance when working with real humans. We focus on settings where a human and an AI agent must coordinate their goals, such as by completing complementary subtasks to achieve an overall objective. We formulate the human-AI collaboration environment as a multi-player *goal-oriented* Markov decision process (MDP).

To enable our investigation, we develop models of human behavior and beliefs by extending the level-$k$ reasoning framework (Stahl II & Wilson, 1994) to account for suboptimal human behavior. We define a *behavioral level-0* model that captures human actions without considering the other agent, allowing for more realistic, non-optimal behavior. Building on this, we introduce a *behavioral level-1* model that captures human beliefs about AI intentions by assuming that humans interpret the AI's actions as if the AI follows the behavioral level-0 model. These models serve as the foundation for our AI design. More specifically, we first use them to train explicable AI policies, policies with actions that are easier for humans to interpret, thereby validating the belief model and demonstrating its utility in shaping AI behavior. We then train collaborative AI agents that explicitly incorporate these models to optimize team performance, using self-play with the behavioral and belief models guiding the AI training.

We validate our approach through extensive human-subject experiments. These experiments evaluate both the explicability of our AI policies, measuring how accurately humans can infer AI intentions, and the collaborative performance of AI agents trained with belief models. Results show that our explicable AI policies produce behaviors that make AI goals easier for humans to infer. Moreover, AI agents trained with models of human beliefs significantly improve team performance, demonstrating the effectiveness of belief-informed AI in enhancing human–AI collaboration.

**Contributions.** The main contributions of our work are summarized as follows:

- We develop a model of human beliefs about AI intentions by extending the level-$k$ framework with a data-driven behavioral level-0 model. This allows the belief model to account for suboptimal human behavior when predicting how humans infer AI intentions from observed actions.

- We use this belief model to guide the design of AI behavior in two settings. First, we train explicable AI policies that choose actions from which humans can more easily infer the AI's intended goal. Second, we train collaborative AI agents that account for both human behavior and human belief formation when optimizing human-AI team performance.

- Through simulations and extensive human-subject experiments, we demonstrate that (1) our belief model better captures human inferences about AI intentions and can be used to design AI policies whose intentions are easier for humans to infer, and (2) our belief-aware collaborative AI agents outperform agents that do not account for human beliefs when paired with real humans. These results underscore the importance of incorporating human beliefs into collaborative AI and highlight the effectiveness of our approach.

## 2  Related Work

Our work contributes to the growing body of research on human-AI collaboration (Carroll et al., 2019; Jaderberg et al., 2019; Bansal et al., 2021; Wilder et al., 2021; Myers et al., 2024). For example, Bansal et al. (2021) demonstrate that optimizing AI in isolation may lead to suboptimal performance in human-AI collaboration, compared to training AI with consideration for human performance. Wilder et al. (2021) show

that training AI systems to complement humans by performing better in areas where humans struggle results in improved collaborative performance. Carroll et al. (2019) illustrate that incorporating a human model, learned from human data, into the training of AI enhances performance when these AI systems collaborate with real humans. Our work extends this line of research by not only accounting for human behavior when designing AI but also incorporating human beliefs about AI intentions.

While the approaches in the above works often assume knowledge or data of humans, there have also been zero-shot coordination techniques being proposed to train cooperative AI agents when human data are not available. For example, Jaderberg et al. (2019) use population-based reinforcement learning to improve the robustness of trained AI agents. Cui et al. (2021) propose the Other-Play algorithm to address the limitations of self-play by leveraging environment symmetries, showing improved zero-shot coordination with both AI and human agents in Hanabi games. Strouse et al. (2021) develop a variety of self-play agents by varying random seeds of neural network initialization and various checkpoints, and train a best response model tailored to the agent population, discovering that humans prefer to partner with this model. Yan et al. (2023) further adapt a mixture of random policies in population-based training, which significantly improves training efficiency while maintaining strong performance in zero-shot coordination in Overcooked games. Zhao et al. (2023) utilize the advances of maximum entropy reinforcement learning to encourage the diversity of agent populations and show the learned AI models could adapt to real humans. This line of work aims to design collaborative AI that is robust to diverse human behavior. However, they do not explicitly account for human beliefs over AI intentions during human-AI collaboration. In this work, we have included these approaches as baselines and show that accounting for human beliefs leads to better performance in our setting.

At a high level, our work aligns with the broader effort to understand and model humans in computational systems (Choudhury et al., 2019; Shah et al., 2019; Kwon et al., 2020; Chan et al., 2021; Reddy et al., 2021; Narayanan et al., 2022; 2023; Treiman et al., 2023; 2024; 2025). More technically, for example, inverse reinforcement learning(Ng et al., 2000; Abbeel & Ng, 2004; Ramachandran & Amir, 2007) aims to infer reward functions in Markov decision processes (MDPs) by observing demonstrations of the optimal policy. When the demonstrator is human, these demonstrations may be noisy or exhibit behavioral biases. Several studies (Evans et al., 2016; Shah et al., 2019; Hughes et al., 2020; Zhi-Xuan et al., 2020) have addressed this by incorporating human behavioral biases into the inference process, aiming to infer both the rewards and biases simultaneously. Imitation Learning (Hussein et al., 2017) also seeks to develop models that can mimic human behavior based on demonstrations.

From the perspective of modeling human beliefs about others' intentions, this has been extensively studied in economics through level-$k$ reasoning (Stahl II & Wilson, 1994; Doshi et al., 2010; Gill & Prowse, 2016) and in cognitive science via Theory of Mind (ToM) (Premack & Woodruff, 1978; Rabinowitz et al., 2018; Chen et al., 2021; Westby & Riedl, 2023; Li et al., 2023). Prior work in behavioral game theory suggests that human behavior in unrepeated games is often better captured by low-level iterative reasoning models, such as level-k and related variants, than by full-rationality assumptions (Wright & Leyton-Brown, 2017). Albrecht & Stone (2018) survey belief/intent modeling in agents, distinguishing between stationary and adaptive behavior representations. Recent work extends ToM modeling to modern AI systems, particularly in multi-agent and large language model settings. In these approaches, agents are equipped with explicit or implicit belief representations to reason about others' intentions, for example through structured ToM benchmarks and multi-agent architectures built on large language models (Chen et al., 2024; Shi et al., 2025; Zhang et al., 2025).

Our work extends this line of research in two ways. First, our belief inference framework differs from previous theory of mind literature by modeling and incorporating human beliefs about AI intentions specifically, using a more realistic human model as the basis for inferring these intentions. Second, and more importantly, beyond modeling human beliefs over others' intentions, our work focuses on the design of collaborative AI agents that account for human beliefs and evaluates the design using human-subject experiments.

## 3 Problem Formulation and Methods

In this section, we formulate the human-AI collaboration problem as a multi-player goal-oriented Markov decision process (Section 3.1). We then introduce our models of human behavior and human beliefs about

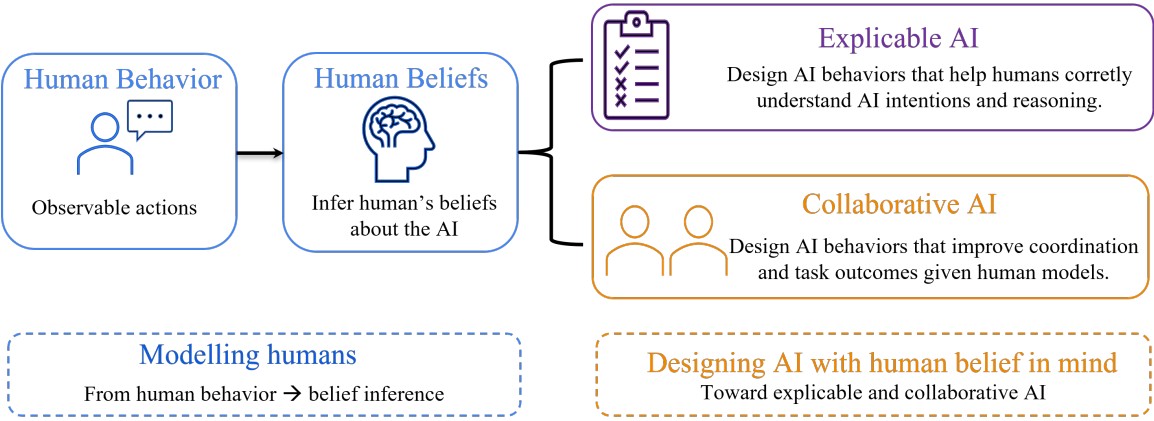

Figure 1: Overview of our human-AI collaboration framework. We first model how humans form beliefs about AI behavior from observed actions. Building on this, we design human-aware AI systems that account for such belief formation, enabling both explicable AI (improving interpretability of AI behavior) and collaborative AI (improving coordination between humans and AI).

AI intentions, followed by methods for incorporating these models into collaborative AI design. Section 3.2 extends the level-$k$ framework to account for suboptimal human behavior when modeling human beliefs. Building on these belief models, Section 3.3 presents methods for designing explicable AI policies that make AI intentions easier for humans to infer from observed behavior. Finally, Section 3.4 introduces collaborative AI agents that explicitly reason about both human behavior and human beliefs to improve team performance.

Figure 1 summarizes the overall framework. We first model human behavior from observed actions, use this behavior model to infer how humans form beliefs about AI intentions, and then use these belief models to guide AI design. This framework supports two complementary design objectives: generating explicable AI behavior that makes AI intentions easier to infer, and training collaborative AI agents that improve coordination with human partners.

## 3.1 Setting

**Decision-making environments.** We formulate the human-AI cooperative decision-making environment as a multi-agent Markov decision process (MDP), represented by $W = \langle S, \alpha, A_i|_{i \in [\alpha]}, R, P \rangle$, where $S$ is a finite set of states, $\alpha$ is a finite set of players, $A_i$ is the action set available to player $i$, $P : S \times A_1 \times \cdots \times A_{|\alpha|} \times S \to [0,1]$ is the transition function that determines the next state given all players' joint actions, and $R : S \times A_1 \times \cdots \times A_{|\alpha|} \to \mathbb{R}$ is the joint reward function assigned to all players given their joint actions.

Note that in this work, we consider the cooperative setting. Therefore, our formulation only incorporates a single reward function $R$ for all players, though this can be easily extended. Moreover, while our formulation could address cases with multiple human and AI players, in this work, we also focus on a two-player cooperative game, i.e., $\alpha = \{1, 2\}$, where one player is a human and the other is an AI agent. During decision making, neither the human nor the AI knows the other player's next action or future plans, and they cannot communicate directly. However, they can observe each other's past actions, allowing them to infer the intentions of the other player and adjust their own actions accordingly.

**Agent intentions.** In this work, we focus on a setting with a goal-oriented MDP, where there is a set of goals $G = \{g_1, \ldots, g_k\} \subseteq S$. The decision-making agent only receives rewards when arriving at one of the goal states, and the goal states are terminal states, i.e., $P(g|g, a) = 1 \ \forall g \in G, a \in A$. Both humans and AI agents can observe past actions but not the specific goals the other agents are trying to achieve. We use this formulation to define an agent's intention in practical terms, that is, an agent's intention refers to the specific goal the agent is aiming to achieve.

### 3.2 Modeling Human Beliefs about AI Intention

Our model extends the level-$k$ framework (Stahl II & Wilson, 1994) in economics. In particular, We start by modeling human behavior by considering humans as level-0 agents who do not account for others' behavior. Unlike prior literature, we address the realistic scenario where level-0 agents may not behave optimally, and we call this model *behavioral level*-0 agents. Then, to model human beliefs about others' intentions,[1] we model humans as an extension of level-1 agents, who assume other agents are *behavioral* level-0 agents[2] and update their beliefs about AI intentions in a Bayesian manner based on the observation of others' behavior. More details are described below.

**Modeling human behavior.** We first describe our models of human behavior under the assumption that humans do not consider other players in the environment (or that they view other players as part of the environment without strategically responding). Our approach follows standard approaches in the literature and primarily serves as a foundation for developing models of human beliefs. In particular, a human behavior model can be represented as $H : W \to \Pi$, mapping a given environment $w \in W$ to a policy $\pi = H(w)$. We provide two examples of human models used in our work below.

- *Standard model.* Consider the standard model in MDPs, in which the goal of the human is to maximize the expected cumulative reward, and their policy only depends on the current state. The model can be represented by $\pi(a|s)$, indicating the probability of choosing action $a$ at state $s$. For the standard model that assumes decision optimality, humans choose actions maximizing the Q-function, where $Q(s, a)$ indicates the expected cumulative reward if the player takes action $a$ in state $s$ and follows policy $\pi$. $Q(s, a)$ could be calculated by standard reinforcement learning techniques such as value iteration or Q-learning.

- *Behavioral level*-0 *model.* We also consider the case where we can learn human behavioral models from their historical behavior using behavioral cloning (Pomerleau, 1988; Torabi et al., 2018), an imitation learning approach that learns a policy from human demonstrations by mapping states to actions using supervised learning methods (Bain & Sammut, 1995). We build a fully connected neural network, where the input is the state encoding and the output is the probability distribution over the action space. The model is trained using the standard gradient descent method with a cross-entropy loss.

**Modeling human beliefs.** We now describe how we develop models for human beliefs about AI intentions. As summarized earlier, our belief models extend the ideas of level-$k$ reasoning and assume that human decision-makers perceive other agents in the environment as behavioral level-0 agents. In our discussion, we also refer to this belief model as *behavioral level*-1 *agents*.

More specifically, we model the human belief updating as a Bayesian inference process, which can be represented as in Equation 1, where $\lambda(g)$ represents the human's prior belief distributions of AI intentions, and $Pr(s_t, a_t|g)$ denotes the probability of observing $(s_t, a_t)$ given the intention (goal) $g$, according to the policy model $\pi(a|s, g)$. This policy model represents how humans perceive the actions of other players. If a human believes the other agent is following the standard model, then the policy is derived from the optimal policy. On the other hand, if humans believe the agent follows the behavioral cloning model, then the policy will be the output of the human model learned from data.

$$
\begin{aligned}
B(g|(s, a)_{1:t}) &\propto \lambda(g) Pr((s, a)_{1:t}|g) \\
&= \lambda(g) \prod_{1:t} Pr(s_i, a_i|g) \\
&= \lambda(g) \prod_{1:t} \pi(a_i|s_i, g) P(s_i|s_{i-1}, a_{i-1})
\end{aligned}
\tag{1}
$$

### 3.3 Designing Explicable AI Policy

Having introduced our model of human beliefs about AI intentions, we now consider how these belief models can guide AI design. We begin with the design of *explicable* AI policies, namely policies that generate

---

[1] In this work, we sometimes use *human beliefs* to refer to humans' beliefs about the intention of the AI agent.

[2] Our model can iteratively extend to higher level-$k$ agents; however, we focus on the case with $k \leq 1$.

behaviors from which humans can more accurately infer the AI's intentions.[3] This setting serves two purposes. First, it provides a direct way to evaluate whether the belief model captures how humans interpret AI behavior. Second, it demonstrates how belief models can be used to shape AI actions, laying the groundwork for training collaborative AI agents in Section 3.4.

Prior work has shown that humans often find it challenging to infer the intentions of other agents from behavioral traces (Ullman et al., 2009; Baker et al., 2011; Zhi-Xuan et al., 2020), a pattern that also appears in our human-subject experiments (see Section 4 and the appendix for additional results). We therefore ask whether an AI agent can choose actions that make its intended goal easier for humans to infer. To do so, we use the belief model developed in Section 3.2 as a proxy for the human inference process during training.

Specifically, we train the AI policy to optimize both task performance and explicability. In addition to the standard MDP reward, we add an explicability bonus based on the belief model. When the goal inferred by the belief model matches the AI's intended goal, the agent receives an additional reward proportional to the log likelihood assigned by the belief model, $\log(\pi(a \mid s, g))$. This objective encourages the AI policy not only to reach its goal, but also to choose actions that make the goal more inferable to humans according to the belief model.

### 3.4 Designing Collaborative AI Agents

Having shown how belief models can be used to design explicable AI policies, we now turn to collaborative AI agents that explicitly incorporate human behavior and belief models to optimize team performance. In the experiments, we illustrate the effectiveness of incorporating models of human beliefs by examining the performance of AI agents paired with humans through simulations and real-world human-subject experiments.

**Training methodology.** The main idea of our training method is through self-play. Our training method uses simulated interaction with different human models. For each AI design, we instantiate a simulated teammate according to the corresponding human model, such as an optimal agent, a behavioral level-0 agent, or a behavioral level-1 agent that updates beliefs about AI intentions. We then train the AI agent with proximal policy optimization (PPO) to maximize team reward when paired with that simulated teammate.

**Collaborative AI agents.** We designed several AI agents based on different assumptions of the human counterpart.

- *Assuming humans are optimal.* We train an AI agent that learns to collaborate with itself through self-play. This is equivalent to assuming the human is acting optimally.

- *Assuming humans are behavioral level-*0*.* We train an AI agent that assumes the human is a behavioral level-0 agent, using behavioral cloning to learn the human's behavioral model on historical data.

- *Incorporating models of human beliefs.* Finally we also train an AI agent that incorporates both the models of human behavior and beliefs into the design of AI agents.

In addition to the AI agents trained using our methodologies, we also include two state-of-the-art baselines, Fictitious Co-Play (Strouse et al., 2021) and Maximum Entropy Population-based Training (Zhao et al., 2023), for comparison in our experiments. Both algorithms develop a single best response model in response to a population of agents with diverse behaviors, thus enhancing the robustness and generality of the learned best response model. The details of the setup and implementations are provided in the appendix.

## 4 Human-Subject Experiments

To evaluate our approach, we conducted multiple sets of human-subject experiments with participants recruited from Amazon Mechanical Turk (MTurk). These experiments evaluate both components of our framework: whether the proposed belief models can support the design of explicable AI policies, and whether

---

[3]Prior research has introduced related concepts, such as legible motion (Dragan et al., 2013; Mavrogiannis et al., 2018) and explicable policy (Chakraborti et al., 2019; Devidze et al., 2021; Gong & Zhang, 2022). We use the term *explicable* to emphasize our focus on policies that reveal AI intentions through behavior.

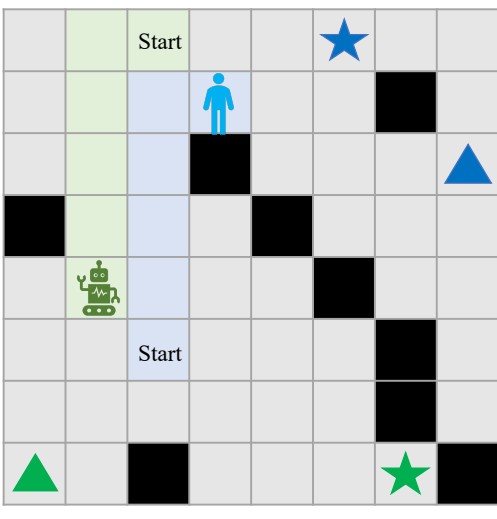
(a) Evaluating explicable AI.

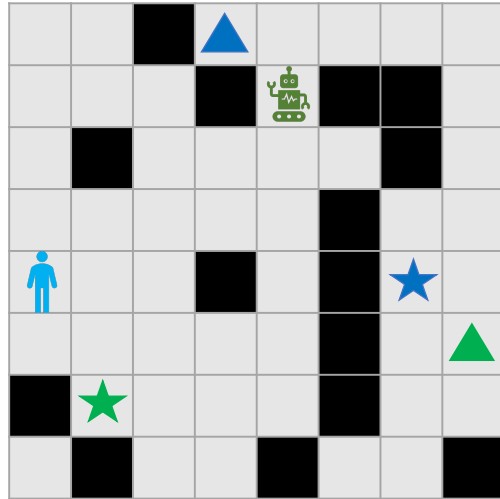
(b) Evaluating collaborative AI.

Figure 2: Interface for the human-subject experiments.

belief-aware collaborative AI agents improve team performance with real humans. Due to space constraints, we present the main experiments and key results in this section, and provide additional experiments, including results from a different environment setup and qualitative analyses of when the approach is most effective, in Appendix A. Table 6 provides a breakdown of all human-subject experiments.

All experiments were approved by our institution's IRB. Workers received a \$1 base payment with the potential for bonus payments, resulting in an average hourly rate of approximately \$14.

### 4.1 Experiments Overview

The main purpose of our experiments is to demonstrate that our proposed methodologies by incorporating human beliefs about AI intentions into the design of AI agents can improve the performance of human-AI collaboration with *real-world human participants*.

**Experiment environment.** Our experiments are conducted in a grid world environment with two players and multiple goals.[4] Specifically, as shown in Figure 2, the grid world is 8 by 8 in size and contains the positions of both players and four possible goals. The players can choose to move {*Up, Down, Right, Left*} or stay in their current grid. They cannot directly communicate but can see each other's positions and act simultaneously. Each player can navigate to two of the four goals: the human player can reach one of the two blue goals, and the AI player can reach one of the green goals. When both agents reach the same type of goal (e.g., both a "star" or both a "triangle"), they earn positive points. They do not earn points if they reach different types of goals or collide (move into the same position). The environments are randomly generated in the experiments. The maximum number of actions is set to 20.

**Experiment descriptions.** Our evaluation consists of a behavior data collection phase followed by two human-subject experiments. We begin by collecting human behavioral data to develop models of human behavior and beliefs, and then evaluate the effectiveness of our explicable and collaborative AI designs.

In the data collection phase (Section 4.2), we recruit participants to interact in the grid-world environment to collect behavioral traces. These traces are used to train a behavioral cloning model of human actions. We then evaluate the predictive accuracy of this learned model on held-out data to assess how well it captures human decision-making patterns.

---

[4]We conducted experiments in two variations of the environments. Since the results are qualitatively similar, we leave the results and discussion for the other variant in the appendix.

In Experiment 1 (Section 4.3), we examine whether the learned belief models can be leveraged to design explicable AI policies, as described in Section 3.3. Participants are shown behavioral traces generated by different AI policies and asked to infer the AI agent's goal. We evaluate whether AI policies trained to account for human beliefs produce behaviors that make AI intentions easier for humans to infer.

In Experiment 2 (Section 4.4), we evaluate our collaborative AI design. We recruit participants, pair them with different AI agents as described in Section 3.4, and measure overall team performance. This experiment examines whether incorporating human belief models into collaborative AI leads to improved human-AI team performance.

Detailed descriptions of the procedures and additional results, including experiments conducted in a different environment setup, are provided in the appendix.

## 4.2 Behavior Data Collection

To develop our belief model and collaborative AI design, we first collect human behavior data in our environments and use behavioral cloning to train the model of human behavior. This behavior model is then used to develop our belief model and collaborative AI.

**Data collection procedure.** We recruited 190 workers from Amazon Mechanical Turk. Each participant was asked to play 30 navigation games within a grid world (as shown in Figure 2) alongside an AI model. The goal of this experiment was to develop a model of human behavior that predicts a sequence of actions given a specified goal. To this end, we instructed participants on the goal to reach in each round and recorded their behavior as the collected dataset.

**Examining the behavior model.** We divided the collected human action data into training (152 workers, 136,000 decisions), validation, and testing sets (19 workers each, 17,000 decisions per set). A 4-layer MLP model was trained to predict the next human action based on the current environment layout. We compared the performance of our learned model with a model based on optimal agent behavior, defined as taking the shortest path to the goal. Training, validation, and test accuracies for both models are shown in Table 1. The results clearly show that human behavior deviates significantly from assumed optimality, underscoring the importance of incorporating realistic human behavior models into human-AI cooperation frameworks. Moreover, the data-driven model trained via behavioral cloning achieves much higher and reasonable predictive performance.

Table 1: The prediction accuracy for human behavior was evaluated assuming optimal behavior and using a data-driven model.

|  | Training Accuracy | Validation Accuracy | Testing Accuracy |
|---|---|---|---|
| Assuming Optimal Behavior | 0.4498 | 0.4327 | 0.4459 |
| Data-Driven Model | 0.8547 | 0.7831 | 0.7899 |

## 4.3 Experiment 1: Evaluating Explicable AI

We now evaluate our proposed explicable AI. Specifically, we examine whether the human belief model can enable us to develop explicable AI policy, as described in Section 3.3, making it easier for humans to infer AI intention based on AI actions. [5]

**Experiment setup.** In this experiment, each recruited participant is presented with the behavioral traces from both agents in the environment, as shown in Figure 2. Each behavioral trace is a sequence of actions

---

[5] We have also directly tested the accuracy of our belief model in inferring human beliefs by comparing its predictions with direct solicitations of users' inferences about others' behavior. The results, included in Appendix A.3, suggest that our model is more accurate in predicting human beliefs compared to the baselines. However, they also demonstrate that human beliefs are generally quite noisy. If we randomly draw a behavior trace from another agent and ask humans to predict the goal, human prediction accuracy is close to random guessing. Therefore, we shift our focus of this experiment to whether AI agents can design their action plans to make it easier for humans to infer the goal.

generated by an AI policy for a player. Participants are then asked to infer the intention of each agent, i.e., the goal each agent is trying to reach. If participants can better infer the goal from the behavior generated by a certain AI policy, it indicates that the AI policy is more *explicable* in that it generates actions that make it easier for humans to infer AI intentions.

**AI policy.** We now describe the set of AI policies that are used for evaluations. As baselines, the first two AI policies are not intended to be explicable.

- *Inexplicable-Optimal-L0*: This AI takes the shortest path towards the goal.
- *Inexplicable-Behavioral-L0*: This AI follows the behavioral model learned from human data.

We then include two AI policies that take actions intended to be explicable. To design explicable AI policies, we need to account for human belief models in the design of AI policies, as the AI will try to take actions that make it easier for humans to infer the goal, assuming models of human beliefs that describe how humans make inferences. The two belief models we account for are: (1) the classical level-1 model, where the human assumes the other agent is taking optimal level-0 actions, and (2) the behavioral level-1 model, where the human assumes the other agent is taking behavioral level-0 actions as described in Section 3.2. Coupled with these two belief models, and using the methodology described in Section 3.3, we train the following two explicable AI policies.

- *Explicable-Optimal-L1*: This AI policy is trained to be explicable but assumes that humans belief model is the classical level-1 model, where they assume the other agent is optimal.
- *Explicable-Behavioral-L1*: This AI policy is trained to be explicable but assumes that human belief model is the behavioral level-1 model, where they assume the other agent is behavioral level-0.

**Experiment results.** We recruited 300 workers from Amazon Mechanical Turk, randomly assigning them to one of four treatments, where each treatment represents showing participants traces generated by one of the AI policies above. Each participant was tasked with identifying the goals of the player in 30 different scenarios. The length of actions is drawn from the range 4 to 8. To incentivize effort, participants were awarded a $0.03 bonus for each correctly identified goal.

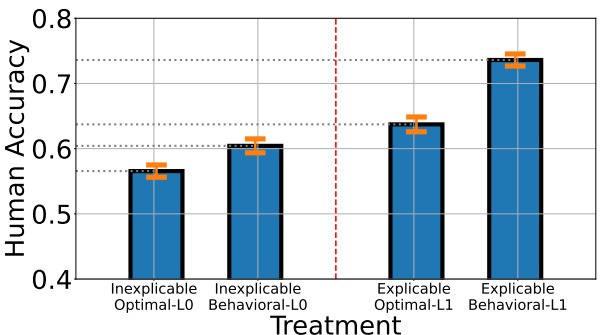

The experiment results are shown in Figure 3. First, we observe that when the AI is not attempting to be explicable, humans generally struggle to infer AI intentions, with the accuracy of correctly inferring the goal for both inexplicable policies being around 55% to 60%, only slightly better than random guessing. However, if we use our methodology from Section 3.3 to train the AI policy to be explicable, it significantly improves the accuracy of humans in inferring AI intentions. Moreover, when the training is coupled with our belief models, the accuracy for humans to

Figure 3: The results of the human-subject experiments in Experiment 1 show that when using our approach to design explicable AI, the accuracy for humans to infer AI intentions is higher than with other AI policies. Moreover, when coupled with our belief model, it results in an AI policy that makes it the easiest to infer AI intentions.

infer AI intentions is much higher than all other policies, demonstrating the effectiveness of both our belief models and the methodologies in training explicable AI policy.

### 4.4 Experiment 2: Evaluating Collaborative AI

As the main experiment, we evaluate our collaborative AI design. The objective is to show that incorporating human beliefs into the design of collaborative AI can enhance human-AI collaborative performance through both simulations and human-subject experiments.

**Collaborative AI design.** As described in Section 3.4, we train collaborative AI agents using three different corresponding human models.

- *Self-Play*: the collaborative AI agent that is trained assuming they are playing with itself.

- *Behavior-AI*: the collaborative AI agent that is trained assuming the human partner is the behavioral level-0 agent. This corresponds to the existing approaches of accounting for human behavior in AI design Carroll et al. (2019).

- *Belief-AI*: the collaborative AI agent that is trained assuming the human partner is the behavioral level-1 agent, i.e., it accounts for human belief models into the design of AI.

In addition to the three collaborative AI agents trained using our methodology, we also include two state-of-the-art baselines for comparison. These methods leverage zero-shot population-based training (PBT) techniques. While they do not explicitly account for specific models of human behavior, they are designed to be robust in collaborating with different teammate behaviors.

- *FCP*: Fictitious Co-Play is a variant of fictitious play proposed by Strouse et al. (2021). It consists of two stages: in the first stage, a population of agents is trained independently using different random seeds and checkpoints; in the second stage, a model is trained as the best response to the agent population. In our experiment, we select the model trained with 8 partners.

- *MEP*: Maximum Entropy Population-based training is proposed by Zhao et al. (2023). The agent population is trained using population entropy bonus to increase the diversity of agents, then a single best response model is trained with respect to the agent population. In our experiment, we select the model trained with 8 partners.

This set of comparisons spans a broad range of state-of-the-art baselines in the collaborative AI literature, including standard multi-agent RL approaches (e.g., self-play), methods that account for human behavior (Carroll et al., 2019) (Behavior-AI), and zero-shot population-based approaches (*FCP* and *MEP*). Our approach is distinct in that it incorporates a model of human belief into the design of collaborative AI.

**Simulations.** We first run simulations to evaluate the performance of different pairings of human models and collaborative AI designs, where we assume the human agent is optimal (*self-play*), a behavioral level-0 agent (*Behavioral Model*), and a behavioral level-1 agent (*Behavior&Belief*). The evaluation is based on $10,000$ randomly generated environments which filter out cases where the distance between the two goals for the same player is smaller than 3 to allow models to adjust their behavior based on the inference of their teammate's actions.

Table 2: Simulation results of collaborative performance over 10,000 testing cases in Experiment 2. The column players represent different AI agents, and the row players represent different simulated human models. The results demonstrate that while the zero-shot methods (FCP and MEP) show stable performance, when human agents infer AI intentions to act accordingly, designing AI that accounts for human beliefs brings significant performance improvements.

| Human Model | AI Agent | | | | |
|---|---|---|---|---|---|
| | Self-Play | FCP | MEP | Behavior-AI | Belief-AI |
| Self-play | **0.6245** | 0.6136 | 0.6131 | 0.5250 | 0.6177 |
| Behavior Model | 0.4902 | 0.5782 | 0.5927 | **0.6334** | 0.5755 |
| Behavior&Belief | 0.6411 | 0.6255 | 0.6484 | 0.6574 | **0.7675** |

We partner the collaborative AI agents with the three simulated human agents and measure the collaborative performance. The simulation results are shown in Table 2. For both the FCP and MEP baselines, despite the absence of human data in their training processes, they outperform the self-play AI when paired with the human behavior model, demonstrating their adaptability to working with teammates exhibiting unknown behavior. However, we also observe that incorporating the appropriate human behavior model directly into the training process can further enhance human-AI collaborative performance, as demonstrated by the superior performance of Behavior-AI when paired with the behavior model. Finally, when working with a

behavioral level-1 agent that attempts to infer AI intentions, our proposed Belief-AI stands out, significantly outperforming other AI agents.

These results highlight the importance of incorporating appropriate human models (of behavior and beliefs) for training collaborative AI. Furthermore, under conditions where humans attempt to infer AI intentions and act accordingly, designing AI that accounts for human beliefs can significantly improve human-AI collaborative performance.

**Human-subject experiments.** While the simulation results are promising, the performance improvement from incorporating human beliefs into AI design is based on the assumption that humans follow our belief model. To examine whether this performance improvement carries over to collaborations with real humans, we recruited 400 workers and randomly assigned them into five treatment groups, each interacting with a different AI design. Participants were tasked with playing 30 games, preceded by three tutorial games designed to familiarize them with the gameplay. Participants could receive a \$0.05 bonus payment for reaching the same type of goal as the AI agent in each of the 30 tasks.

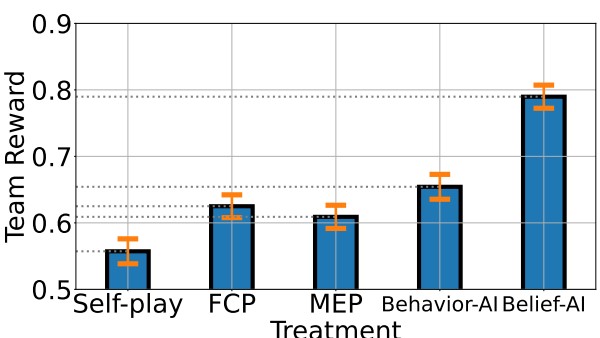

Figure 4: Human-AI collaborative performance in Experiment 2 shows that Belief-AI significantly outperforms all baselines when paired with real humans.

We first examine whether the data-driven model learned during behavior data collection still makes good prediction on human behavior in Experiment 2 (i.e., whether the learned behavior model performance generalizeds to new experiment). Overall, we find that the model achieves an accuracy of 73.19% in predicting human behavior in Experiment 2 across all treatments, demonstrating the generalizability of the learned human behavior model. More detailed discussion is provided in Appendix A.5.

We then examine the collaborative performance for pairing human participants with different AI designs. The experiment results are shown in Figure 4. They demonstrate that the collaborative AI trained with models of human beliefs achieved the highest collaborative performance when working with real humans, significantly outperforming other AI designs. This outcome not only validates our AI design but also suggests that real human actions align with our belief models in this environment.

More formally, statistical analysis shows significant differences in the performance of AI models paired with humans, with $p < 0.0001$ for comparisons between self-play AI and Behavior-AI, and $p < 0.0001$ for comparisons between Behavior-AI versus Belief-AI. The differences between MEP and FCP, as well as between FCP and Behavior-AI, are not significant ($p > 0.1$). The difference between self-play AI and any other treatments is significant ($p < 0.0001$).

## 5 Conclusion and Discussion

Our work investigates whether accounting for human beliefs about AI intentions can improve human-AI collaboration. We make several contributions. First, we develop models of human behavior and beliefs about AI intentions by extending the level-$k$ framework to account for suboptimal human behavior. Second, we use these belief models to guide AI design, both by generating explicable AI behavior that makes intentions easier for humans to infer and by training collaborative AI agents that account for human belief formation. Finally, through extensive human-subject experiments, we show that belief-aware AI policies are more explicable to humans and improve team performance when paired with real-world users. These results demonstrate the value of considering human beliefs in the design of collaborative AI and provide evidence for the effectiveness of both our belief model and agent design. More broadly, our findings suggest that modeling how humans interpret and reason about AI behavior may be critical for building collaborative systems that are not only

effective, but also predictable and easier for humans to coordinate with. Overall, our work points to a promising direction for building AI systems that more effectively coordinate with their human partners.

**Generalization and limitations.** Our approach has proven effective through human-subject experiments conducted in grid-world environments. These controlled settings allow for systematic evaluation and replication, but, as with most human-subject studies, the findings are limited to the specific experimental context. To improve external validity, future work should evaluate the approach in more realistic domains, such as physical robotics or complex multi-agent systems. Our current approach also relies on historical behavioral data to model human actions and belief inferences, implicitly assuming that these patterns remain stable over time. However, this assumption may not hold in real-world scenarios, where human reasoning and adaptation may evolve over prolonged interactions or changing task demands. For example, users may learn to interpret AI behaviors differently over time or adjust their strategies as they gain familiarity with the AI partner. Finally, it would be valuable to examine the scalability of our approach to larger and more diverse populations and task settings, including those with heterogeneous goals, prior experiences, or varying levels of trust in AI. These efforts would help assess the generalizability and practical deployment potential of belief-informed collaborative AI.

# Broader Impact Statement

This work advances the design of collaborative AI systems by explicitly modeling how humans form beliefs about AI intentions and incorporating these beliefs into AI decision-making. A potential positive impact of this research is its ability to improve coordination and efficiency in human-AI teams. By enabling AI systems to act in ways that are more interpretable and consistent with human expectations, our approach may benefit applications such as healthcare decision support, industrial collaboration, and workflow assistance, where effective teamwork between humans and AI is critical. More broadly, this work contributes to the development of human-centered AI systems that account for people's cognitive processes rather than treating human behavior as fixed or exogenous. Beyond performance gains, belief-aware collaborative AI may also support greater transparency and trust. Our explicable policies are designed to make AI intentions easier for humans to infer from observed behavior, which could help reduce confusion and miscoordination in human-AI environments. By explicitly modeling how AI actions shape human beliefs, this work highlights the importance of designing AI systems that communicate their goals implicitly through behavior, complementing existing efforts in explainable and interpretable AI.

At the same time, this line of research also raises ethical considerations. Systems that reason about and influence human beliefs could be misused to manipulate users or steer their behavior in ways that primarily serve the system's objectives rather than the human's interests. While our work focuses on cooperative settings with shared goals, similar techniques could be applied in contexts where incentives are misaligned. To mitigate such risks, future deployments of belief-aware AI should incorporate safeguards such as explicit user consent, transparency about system objectives, and oversight mechanisms that help ensure alignment with human values. Additionally, our experiments were conducted in controlled environments with informed participants and institutional review board approval, and we emphasize that translating these methods to real-world domains requires careful evaluation of user welfare and autonomy. Another limitation concerns generalization. Our models of human behavior and beliefs are learned from specific populations and task settings, and may not transfer directly to more diverse users or complex environments. Overreliance on imperfect human models could exacerbate inequities or lead to degraded performance for underrepresented groups. Addressing this challenge will require broader data collection, continual model updating, and inclusive evaluation practices.

Overall, this work aims to advance collaborative AI in a direction that explicitly accounts for human beliefs and adaptation. We hope it encourages further research on responsible, transparent, and human-centered AI systems that treat people as active partners rather than passive components in sociotechnical systems.

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

# A    Additional Experiments

Besides experiments in the main text, we also conduct additional experiments in a simple grid world environment with multiple goals to illustrate our study of humans' beliefs about AI behavior. We also explore how to improve the model accuracy of behavioral cloning approach.

## A.1    Experiment Environment: Grid World with Multiple Goals

We use grid worlds of size of $6 \times 6$ in both simulations and human experiments. Similar to the environment setup in Section 4.1, the grid world contains a start position, two goal positions, and some blocked positions that the player cannot enter. The player needs to move from the start position towards one of the goal positions. The player can choose to move {Up, Down, Right, Left}. The player will get a positive reward upon reaching the goal, and we set the maximum number of actions to be 20.

We first recruit participants to engage in a single-player game and record their behavior. We then build human behavior model using behavioral cloning. To evaluate our proposed approaches in modeling human beliefs and designing cooperative AI, we conducted additional two sets of experiments. In the second set of experiments, we provide participants with different traces of actions from other agents, again in a single-player game, and ask participants to infer the goal of the agents. This experiment helps us evaluate whether our belief model leads to better predictions of human beliefs over others' behavior. Afterwards, we conduct a two-player game in the third experiment, where humans are paired with different AI agents to examine the team performance of our design of collaborative AI.

The interfaces of our experiments can be seen in Figure 5a, 5b, and 5c. The detailed descriptions of the experiment and interfaces are included in Appendix C.

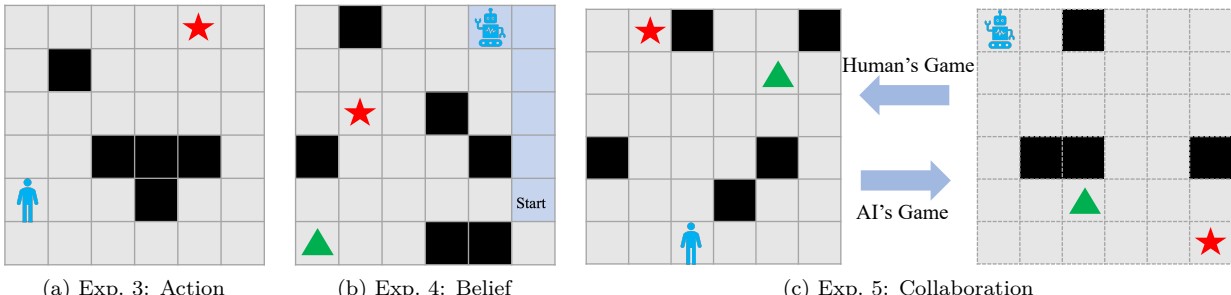

(a) Exp. 3: Action            (b) Exp. 4: Belief                    (c) Exp. 5: Collaboration

Figure 5: Human-subject experiment interfaces. In Experiment 3, each participant is asked to control the player to move to the goal (red star). In Experiment 4, each participant is provided a trace of the behavior by another agent, and is asked to infer which goal the agent is trying to reach. In Experiment 5, each participant is playing with an AI agent in separate environments. The participants only receive bonus rewards by reaching the same goal (red star or green triangle) as the AI agent.

## A.2    Experiment 3: Evaluating Behavior Models in Single-Player MDP

For data collection, we recruited 200 workers from Amazon Mechanical Turk. Each recruited worker was asked to play 15 navigation games within a grid world, as shown in Figure 5a. Our goal is to leverage the collected data to create a data-driven model of human behavior. We divided the collected data of human actions into three sets: training, validation, and testing. The training set comprised data from 160 workers, including approximately 70,000 instances of user decisions, while the validation and testing sets each contained data from 20 workers, amounting to around 8,800 instances of user decisions each.

**Evaluation of data-driven behavior models.** The training, validation, and test accuracies of assuming optimal behavior and behavioral models are presented in Table 3. Results show that data-driven model could predict human behavior more accurately than assuming humans are optimal.

Table 3: The prediction accuracy for human behavior for different human models in Experiment 3.

|  | Training Accuracy | Validation Accuracy | Testing Accuracy |
| --- | --- | --- | --- |
| Assuming Optimal Behavior | 0.7266 | 0.6964 | 0.7131 |
| Data-Driven Model | 0.9189 | 0.8136 | 0.8422 |

### A.3 Experiment 4: Evaluating Explicable AI

We developed human belief models (behavioral level-1) similar to previous experiments. We examine whether belief inference using behavioral model aligns with real humans, and whether we can design AI behavior such that it is easier for humans to infer the goal of the AI agent.

**Experiment setup.** The experiment setup is presented in Figure 5b. The grid world contains a starting position and two goal positions. For each participant, we show them a trace of behavior from another agent and ask the participant to infer which of the two goal the agent is trying to reach.

**Experiment 4.1: Examining the belief models.** We recruited 200 workers from Amazon Mechanical Turk to compare standard level-1 and behavioral level-1 model. Each worker was asked to infer the goal for 25 behavioral traces. We compared the performance of two belief models and the worker accuracy, as shown in Table 4. As we can see from the table, humans are generally poor at inferring the goal of other agents: their accuracy only reaches 59.37% in inferring the goal of the other agent. Moreover, the behavioral level-1 model, which accounts for the human behavior model in the belief model, captures human beliefs better than the standard level-1 model, which assumes human behavior is optimal.

Table 4: Performance comparison between Bayesian inference framework assuming standard model and human behavior model.

|  | Consistency to Human Predictions | Cross Entropy Loss |
| --- | --- | --- |
| Standard level-1 | 0.4977 | 0.9284 |
| Behavioral level-1 | 0.5764 | 0.7506 |
| True goals | 0.5937 | 0.6631 |

**Experiment 4.2: Developing explicable AI policy.** As demonstrated in Experiment 4.1, humans generally struggle to infer the goals of other agents based on others' behavior. To design AI agents which makes inference of their goals easier by humans, we train AI agents to maximize the likelihood that humans can accurately infer these goals from their behavior.

We recruited 400 workers from Amazon Mechanical Turk, randomly assigning them to one of two treatments, to assess the behavior of AI agents. Each participant was tasked with identifying the goals of the player in 30 different scenarios. Participants were awarded a $0.03 bonus for each correctly identified goal. Figure 6 displays the results of human evaluation. The findings indicate that humans achieve higher accuracy in inferring the correct goals of AI agents when employing Behavioral level-1 model.

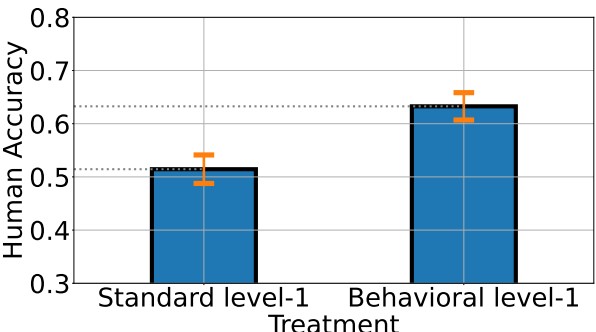

Figure 6: Human evaluations of belief inference accuracy regarding AI goals.

### A.4 Experiment 5: Evaluating Collaborative AI

Utilizing developed models of human behavior and beliefs, we follow the same methodology in Experiment 2: train different collaborative AI, pair them with different human models, and examine the col-

Table 5: Simulation results of human-AI collaborative rewards in Experiment 5. Columns players are different AI agents, and row players are different simulated human models.

| Human Model | AI Agent | | |
|---|---|---|---|
| | Self-play | Behavior-AI | Belief-AI |
| Self-play AI | **0.7828** | 0.4780 | 0.6164 |
| Behavior | 0.5926 | **0.7584** | 0.4552 |
| Behavior&Belief | 0.6919 | 0.7268 | **0.7813** |

laborative performance in simulations and human subject experiments (via recruiting 300 workers).

The setup of our Experiment 5 is shown in Figure 5c. The goal for the human-AI team is for both agents to reach the same goal in their own environments (both reaching "red star" or both reaching "green triangle") within a time limit. The team will not get points if they reach different goals or one of the players fails to reach any goal.

Simulations results are shown in Table 5. Simulations indicate that collaborative performance is highest when an AI agent is paired with the human model used to train it. Figure 7 presents the average collaborative reward in human-subject experiments. Statistical analysis revealed significant differences in the performance of AI models when paired with human participants, with $p$-values of 0.0166 for comparisons between the treatments *self-play* and *Behavior-AI*, and $p < 0.0001$ for *Behavior-AI* versus *Belief-AI*. These results demonstrate that incorporating human beliefs into the deign of AI agents enhances collaborative performance when working with real humans.

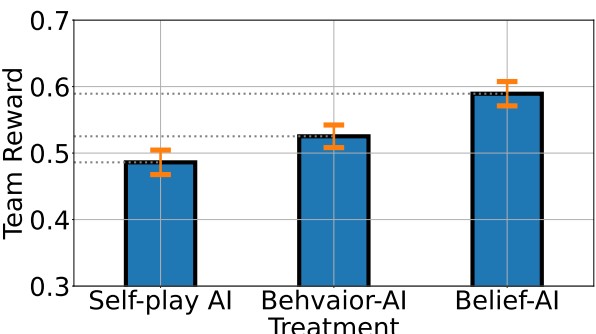

Figure 7: Average collaborative reward in Experiment 5.

### A.5 Extensive Results of Behavioral Cloning

We investigate techniques to enhance the accuracy of behavioral cloning models. Our experiment results reveal that human actions depend significantly on historical actions. We explicitly include the action history of both AI and human players as part of the model input to determine if this improves the prediction of human actions. The findings, presented in Figure 8, indicate that incorporating action history enhances the accuracy of our behavioral cloning model predictions. Additionally, our belief model consistently boosts prediction accuracy across nearly all scenarios.

During data collection phase, we aggregate all data to train a behavioral cloning model. We also train models on human data collected with different AI teammates and evaluate them across treatments, observing consistent average accuracy ranging from 75.4% to 78.9%. Since no significant differences were detected, we aggregate all the data to create a single human model. We also observed suboptimal performance from human participants in this experiment. Even when human participants are provided with suggested goals during data collection, there is only a 55.0% chance that both players will reach the same type of goals across all treatments. Besides, there is a notable chance of two players colliding (about 15.8%) or ending in different types of goals (about 25.6%).

We also evaluate the model's consistency accuracy in Experiment 2 and across different environment layouts. The data-driven model with a random goal achieves an accuracy of 61.15%, while incorporating belief inference improves accuracy to 73.19% across all treatments, consistent with the trend shown in Figure 8. While the environment layout influences model accuracy, on average, the behavioral cloning model aligns with human behavior significantly better than assuming an optimal model. Figure 9 highlights scenarios where

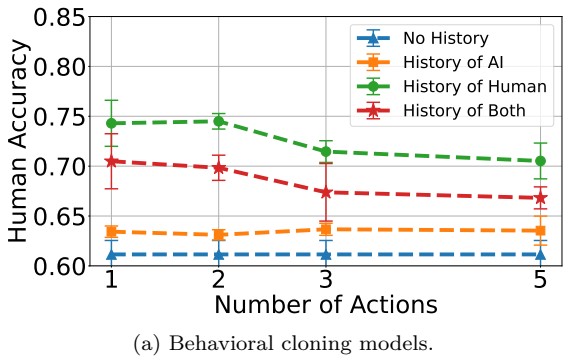

(a) Behavioral cloning models.

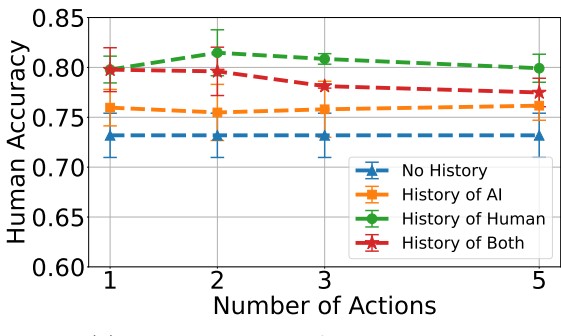

(b) Behavioral cloning & belief models.

Figure 8: Evaluation of model accuracy on human-subject experiment data through incorporation of action history.

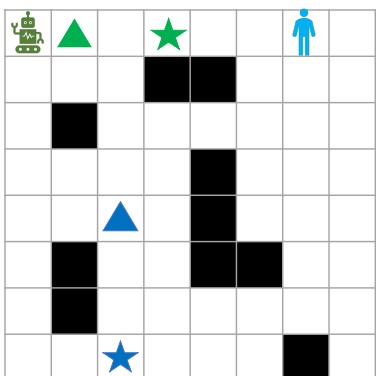

(a) Layout where Belief-AI (11 participants) and Self-play AI (9 participants) perform similarly.

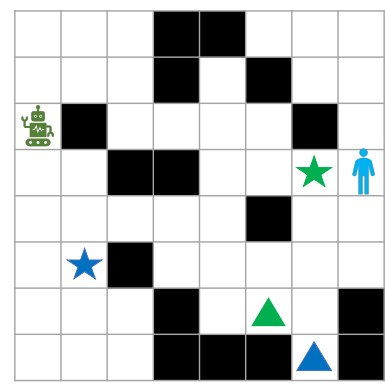

(b) Layout where Belief-AI (12 participants) significantly outperforms Self-play AI (13 participants).

Figure 9: Examples of game layouts highlighting variations in performance across treatments.

performance varies across treatments in Experiment 3. In the layout shown in Figure 9b, most AI types remain in their starting position and act only after the human player shows a clear preference for their goal. Here, the performance of Self-play AI is almost identical to Belief-AI, with no significant differences across other treatments. However, in the layout shown in Figure 9a, Belief-AI moves upward to signal its preference for the star, resulting in 10 out of 12 human players also reaching the star. In contrast, Self-play AI moves downward without clearly indicating its goal, leading to only 5 out of 13 participants aligning with the AI's goal.

## B  Implementation Details

All our experiments are conducted on a cluster of 40 CPU cores (Intel Xeon Gold 6148 CPU @ 2.40GHz), 2 GPUs (NVIDIA Tesla V100 SXM2 32GB), and a maximal memory of 80GB. For completeness, we provide the details of the implementation and the tuning of hyperparameters below.

**State encoding of grid world.** For a grid world of size $N \times N$ and contains $k$ different objects, we use $k$ channels to encode the environment state. Each channel is a binary matrix of size $N \times N$, indicating whether it contains specific objects. For example, in a single-player environment with one goal (environment in Experiment 4), we encode space, blocks, the start position, and goal positions as four channels. Thus the

input state is of size $4 \times 6 \times 6$ before flattening. The state encoding is used as input of our behavioral cloning models and AI agents.

**Behavioral cloning for grid world of size** $6 \times 6$**.** We construct a neural network with four fully connected layers to model human behavior. The input is the state encoding, represented as a matrix of dimensions $4 \times 6 \times 6$, and the output is the probability of taking various actions. The network parameters are initialized using the Glorot uniform initializer. We employ the Adam optimizer and cross-entropy loss to optimize the neural network parameters. The batch size is set to 1024, and the maximum number of training epochs is set to 1000. Training is halted early if the validation loss begins to increase. The hyperparameters are adjusted based on the performance on the validation dataset. The number of nodes for each hidden layer is tuned within the range $\{64, 128, 256, 512, 1024\}$, the initial learning rate within the range $\{0.001, 0.003, 0.01, 0.03, 0.1\}$, and the L2 penalty within the range $\{0, 0.01, 0.03, 0.1\}$. Each training process takes 10 to 30 minutes depends on hyperparameters. In our reported results, we choose hidden layer size of 512, learning rate 0.01 and L2 penalty 0.01.

**Behavioral cloning for grid world of size** $8 \times 8$**.** The neural network structure mirrors that used for the grid world of size $6 \times 6$. The input is the state encoding, represented as a matrix with dimensions $8 \times 8 \times 8$, and the output is the probability of taking various actions. The initialization, optimizer, loss function, batch size and training process are consistent with those used in the grid world of size $6 \times 6$. The maximum of training epochs is set to 3000. The hyperparameters are adjusted based on the performance on the validation dataset. The number of nodes for each hidden layer is tuned within the range $\{1024, 2048, 4096\}$, the initial learning rate within the range $\{0.001, 0.003, 0.01, 0.03, 0.1, 0.3\}$, and the L2 penalty within the range $\{0, 0.01, 0.1\}$. In our reported results, we choose hidden layer size of 2048, learning rate 0.01 and L2 penalty 0.01.

**AI agent training for grid world of size** $6 \times 6$**.** We utilize the Proximal Policy Optimization (PPO) and actor-critic framework for our AI model. Both the actor and critic networks employ four fully connected layers, maintained at the same size. The input is the state encoding of a two-player game, which includes two game states and the index vector (the input vector length is 290). The critic network outputs a single value, while the actor network produces a distribution over the action space. The batch size is set to 256, and the maximum number of training episodes is limited to 1000. The discount factor is set to 0.99. For PPO optimization, the clipping parameter $\epsilon$ is fixed at 0.2, and the number of PPO epochs is 10. The number of nodes for each hidden layer is tuned within the range $\{64, 128, 256, 512, 1024\}$, and the initial learning rate is adjusted within the range of $\{0.001, 0.003, 0.01, 0.03, 0.1\}$. Each training process takes about 2 to 6 hours depends on hyperparameters. In comparisons between the self-play AI and the AI trained to play with a human model, we use the same model size (hidden layer size of 256) and the same initial learning rate of 0.01.

**AI agent training for grid world of size** $8 \times 8$**.** The model structure mirrors that of the grid world of size $6 \times 6$. The input size is adjusted to accommodate the new state representation (the input vector length is 514). The training batch size, discounting factor, maximum number of training epochs and clipping parameters remain unchanged. The number of nodes for each hidden layer is tuned within the range $\{256, 512, 1024, 2048\}$, and the initial learning rate is adjusted within the range of $\{0.01, 0.03, 0.1, 0.3, 0.5\}$. In comparisons between the self-play AI and the AI trained to play with a human model, we consistently use the same model size (hidden layer size of 512) and the same initial learning rate of 0.1.

**Selection of population size.** For FCP and MEP algorithms used in the main text, the population size will directly influence the model performance. We

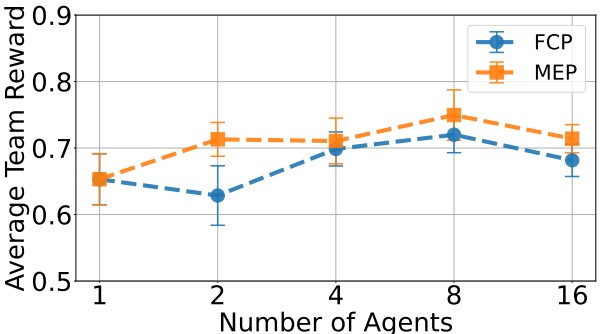

Figure 10: Average team reward for models trained across different population sizes using the FCP and MEP algorithms.

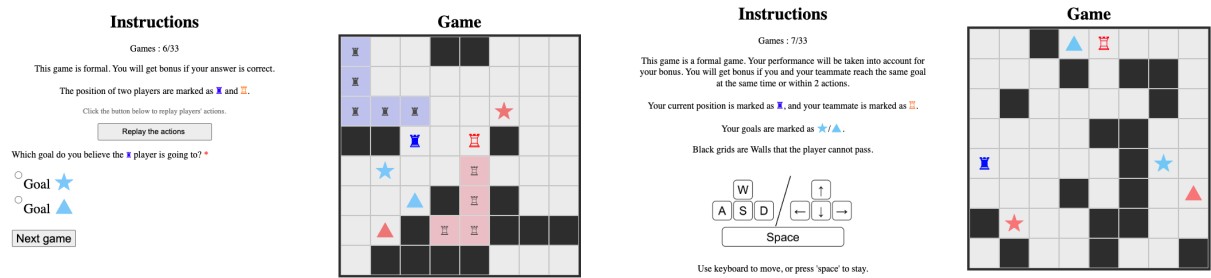

(a) Experiment 1 interface: humans infer goals of other players in grid worlds of size $8 \times 8$.

(b) Experiment 2 interface: humans play games with AI agents in grid worlds of size $8 \times 8$.

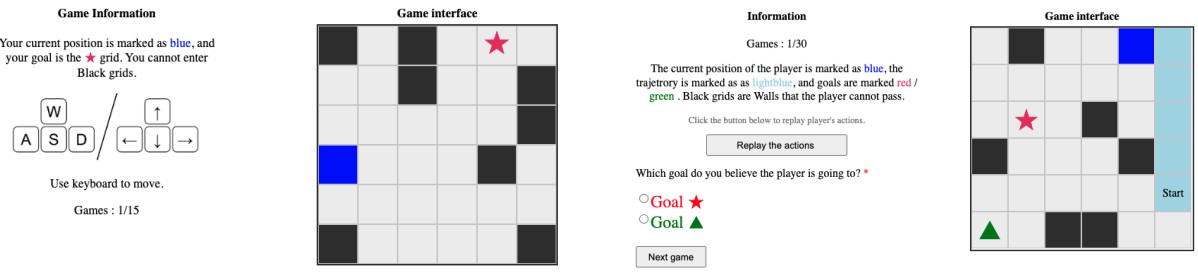

(c) Experiment 3 interface: humans play single-player navigation games in grid worlds of size $6 \times 6$.

(d) Experiment 4 interface: humans infer goals of other players in grid worlds of size $6 \times 6$.

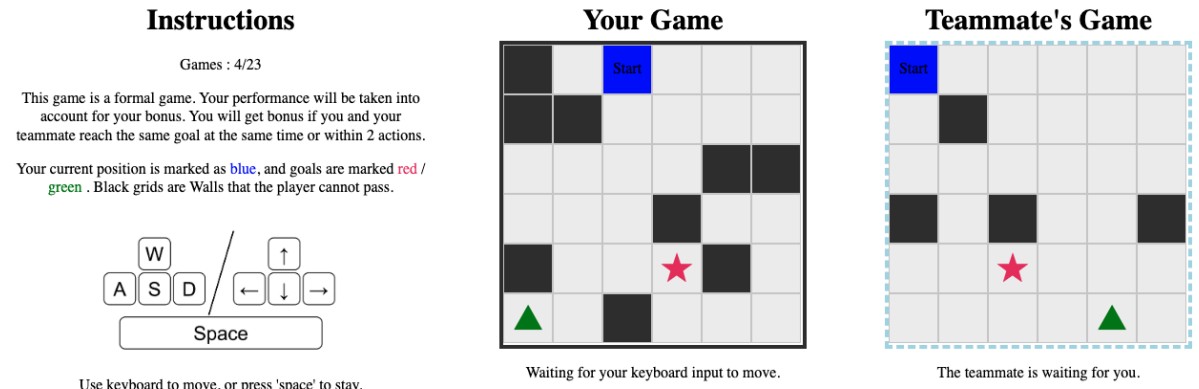

(e) Experiment 5 interface: humans play games with AI agents in grid worlds of size $6 \times 6$.

Figure 11: Human-subject experiment interfaces.

vary the number of agents in our simulations as shown in Figure 10. The model trained with 8 partners is chosen for use in further simulations and human-subject experiments for comparative analysis.

## C   Human Experiment Details

We include more details about human-subject experiments. The experiment interface is shown in Figure 11. We conduct four human-subject experiments, and recruited 1990 workers from Amazon Mechanical Turk in total. Table 6 lists the number of workers, the number of tasks per worker and the number of treatments in each experiment. Table 7 contains the demographic information of all the workers.

Table 6: Number of workers in each human-subject experiment.

|  | # Workers | Tasks per Worker | Treatments | Base Payment | Bonus per Task |
|---|---|---|---|---|---|
| Data Collection Phase | 190 | 30 | 3 | $1.00 | $0.05 |
| Experiment 1 | 300 | 30 | 4 | $1.00 | $0.03 |
| Experiment 2 | 400 | 30 | 5 | $1.00 | $0.05 |
| Experiment 3 | 200 | 15 | 1 | $1.00 | 0 |
| Experiment 4.1 | 200 | 25 | 1 | $1.00 | 0 |
| Experiment 4.2 | 400 | 30 | 2 | $1.00 | $0.03 |
| Experiment 5 | 300 | 20 | 3 | $1.50 | $0.05 |

Table 7: Demographic information of all the participants in our experiments.

| Group | Category | Number |
|---|---|---|
| Age | 20 to 29 | 714 |
|  | 30 to 39 | 974 |
|  | 40 to 49 | 176 |
|  | 50 or older | 72 |
|  | Other | 54 |
| Gender | Female | 639 |
|  | Male | 1319 |
|  | Other | 32 |
| Race / Ethnicity | Caucasian | 1772 |
|  | Black or African-American | 105 |
|  | American Indian/Alaskan Native | 36 |
|  | Asian or Asian-American | 29 |
|  | Spanish/Hispanic | 11 |
|  | Other | 37 |
| Education | High school degree | 66 |
|  | Some college credit, no degree | 22 |
|  | Associate's degree | 36 |
|  | Bachelor's degree | 1457 |
|  | Graduate's degree | 353 |
|  | Other | 56 |

