# OpenReview forum: "Belief‑Aware Collaborative AI: Planning with Human Beliefs of AI Intentions"
_TMLR — Under review for TMLR_

### Review · Reviewer_VVns · 2026-06-03

**Summary Of Contributions:**

In this work, the authors propose to train agents that incorporate and reason about the beliefs of their human collaborators when taking action. The authors argue that their proposed believe aware agents are better to collaborate with, as their actions and goals are easier to predict and understand ('explicable') by reflecting and meeting the expectations of the human collaborator. The authors propose a level-k reasoning loss for modeling beliefs about the collaborator, and infer agent actions that then best match the inferred believe states.

Experiments target simulated self-play, human-subject behavioral cloning and human-AI collaborative settings. Experiments are carried out on a simple randomized grid world under a limited defined set of predefined goals and behaviors. Agents are either trained to expect humans to act according to an optimal policy, learn from behavioral cloning or apply a proposed reflective policy that tries to infer the human's belief of the agent's abilities. Experiments seem to support the claimed efficiency and explicability gains.



**Strengths**

1. We paper is generally well written and structured. Related work addresses works on collaborative settings and AI agents. References to historic works on belief modeling and theory of mind are provided and seem to roughly cover the area.
2. Although acting in a rather simple environment, the experiments and data collection seem to be well setup under a set of clearly stated research questions. The simplicity of the environment allows the extraction and sole analysis of motivations and believe states, and prevents confusion or degradation of agent performances due to difficulties of learning to navigate the environment itself. Experimental results show the well working of the approach under a, small, but clearly defined set of agent behaviors. Results and their interpretation seem to support the proposed collaborative AI approach and clearly answer the posed research questions.
3. Finally, I want to highlight the well framed broader impact statement; discussing benefits of collaborative reflective agents and at the same time highlighting possible ethical concerns on potentially manipulative agents and goal misalignment. Given that all human inferences about agent behavior stem purely from the sole observation of the agent's actions, --without the possibility of introspection into the agent's internal believe states--, manipulative agents and goal misalignment are critical concerns that were addressed adequately.



**Weaknesses**

1. Although the paper is generally well written and motivated, its technical/formal parts are kept rather short with equation (1) being the only equation in the whole paper. While this does not constitute a weakness by itself, I found it quite difficult to sort out which experiments optimize for the various described goals and how those translated to which exact loss/reward functions. Particularly, for the proposed --which I believe to be the most relevant-- 'explicable-behavioral-L1' and 'Belief-AI' agents, a more explicit formalization of the loss/reward function would help to improve the paper.
2. Similar to the previous point, the transition from the second to the third line of Eq. (1), particularly the decomposition of $Pr(s_i,a_i|g)$, is not immediately obvious to me from the preceding text. The authors should explain better how this is decomposition is formally established and/or provide the graphical model of the MDP that gives rise to it.
3. In their paper, the authors focus primarily on a setting that poses no difficulties or constraints on the agents (e.g. every agent is always able to perform the task independently) and behaviors seem to be uniquely identifiable (possibly within the first few actions taken). Given that the authors have constructed a measure to tell policies apart, they should provide further insights into the agent's belief state (or rather the inferred state according to Eq. (1)) throughout the episode to allow for better insights on the complexity and development of the belief state of the agents: e.g, at which point in time and with how much certainty do the agents infer their opposite's believed goal under the presented setup?



**Minor**

* The third line of Eq. (1) uses $P$ instead of $Pr$.
* Similar to the remaining figures, the paper could be slightly improved by including figure 1 as a PDF/vector graphic to improve its render quality and allow text selection.
* Even though described in the text, figure 2's caption could provide a bit more detail on the task that is being shown to make it stand alone a bit better.
* The results shown in the figures should be provided as exact numbers somewhere in the paper.
* Suggestion: The font sizes in Figure 3 vary greatly (also the widths of the bar borders in all figures are a bit too bold for my taste), making the figures slightly unpleasant to look at. Maybe the authors can find a way to improve their presentation.

**Audience:**

Yes

**Audience Explanation:**

Human-AI interaction is a timely topic. Even though the authors demonstrate their approach on a rather simple grid-world, the experiments seem to be well setup and should be easy to extend beyond the shown environment.

**Broader Impact Concerns:**

None. I find the Broader Impact Statement to be very considerate in highlighting the possible risks and opportunities of the presented approach.

**Claims And Evidence:**

Yes

**Claims Explanation:**

The authors clearly state, setup experiments and answer their research questions with the recorded evidence. The interpretation of results seems to align with the shown data.

**Requested Changes:**

The requested changes have mainly been outlined in the weaknesses above. While points 1 and 2 are of a rather technical nature, I deem weakness 3 to be the most critical for showing sufficient complexity of the experiments. As already suggested above, it would be interesting to see the overall development of the belief state according to Eq. (1).

Finally, (and maybe I have missed it somewhere among the various proposed policies): have the authors tested a simple RL training, with human collaboration data as the other agent, but without any belief modeling? In the Behavior-AI / Belief-AI setup, the behavior-AI still seems to perform level-0 belief reasoning, possibly biasing the agent and weakening results?

---

### Review · Reviewer_PJKY · 2026-06-29

**Summary Of Contributions:**

The paper studies human-AI collaboration under the premise that humans adapt their behavior based not only on observed AI actions but also on their beliefs about the AI's intentions. To capture this phenomenon, the authors extend the classical level-k reasoning framework by replacing the standard optimal level-0 agent with a behavioral cloning model learned from human demonstrations. The resulting belief model is then used to design both explicable AI policies that make AI intentions easier for humans to infer and collaborative AI agents that explicitly account for human belief formation during training. The proposed approach is evaluated through simulations and extensive MTurk human-subject experiments, where it demonstrates improvements in intention inference accuracy and collaborative performance over self-play, behavior-aware training, and population-based baselines. The paper addresses an important problem in human-centered AI and provides an interesting integration of behavioral modeling, Bayesian inference, and reinforcement learning.

**Additional Comments:**

Overall, I found the paper well written, clearly motivated, and easy to follow. The inclusion of human-subject experiments substantially strengthens the work compared to many collaborative RL papers. My primary reservation concerns the technical novelty, which appears more incremental than the paper sometimes suggests, and the limited experimental scope beyond simplified grid-world environments. Addressing these limitations would considerably strengthen both the scientific contribution and the practical significance of the work.

**Audience:**

Yes

**Audience Explanation:**

The paper addresses an active research area at the intersection of reinforcement learning, human-AI interaction, multi-agent systems, and cognitive modeling. The idea of explicitly incorporating human beliefs into collaborative AI design is relevant to researchers working on cooperative AI, theory of mind, explainable AI, and human-centered machine learning. Although the methodological novelty is somewhat incremental, the human-subject evaluation and practical motivation make the work likely to be of interest to a broad portion of the TMLR audience.

**Broader Impact Concerns:**

The paper already contains a thoughtful broader impact discussion covering potential misuse through manipulation of human beliefs and emphasizing deployment safeguards. I do not have significant additional ethical concerns beyond those already acknowledged by the authors. However, future work should further examine whether belief-aware systems behave fairly across users with diverse reasoning patterns and whether learned human models introduce biases that could disproportionately affect different user populations.

**Claims And Evidence:**

Yes

**Claims Explanation:**

The experimental evaluation is generally thorough and supports the primary empirical claims within the scope of the environments considered. The authors perform both simulation studies and large-scale human-subject experiments, which strengthens the credibility of the reported improvements. The comparisons against multiple baselines, including self-play, behavior-aware training, and population-based methods, are appropriate and demonstrate consistent gains for the proposed approach. However, some claims regarding the broader significance of modeling human beliefs would benefit from stronger evidence. In particular, the validation of the proposed belief model is largely indirect through downstream task performance rather than direct evaluation of whether the inferred belief distributions accurately match human reasoning. Additionally, all experiments are conducted in relatively simple grid-world environments, making it difficult to assess whether the observed improvements would generalize to more realistic collaborative settings.

**Requested Changes:**

The primary concern is the limited methodological novelty of the proposed belief model. While replacing the classical optimal level-0 agent with a learned behavioral model is a reasonable extension, the underlying Bayesian inference framework remains largely standard. The paper would benefit from a clearer discussion positioning this contribution relative to existing Bayesian intention inference and Theory-of-Mind literature.

The experimental evaluation should also be strengthened. All experiments are performed in relatively small grid-world environments with discrete goals, making it difficult to assess scalability and practical applicability. Additional experiments on more complex environments or tasks would significantly strengthen the paper's impact.

The paper would also benefit from additional ablation studies. In particular, it is unclear how sensitive the collaborative performance is to inaccuracies in the behavioral cloning model or to each component of the proposed framework individually. Understanding these sensitivities would provide greater confidence in the robustness of the approach.

Finally, the validation of the proposed belief model should be made more direct. Since modeling human beliefs is the central contribution of the paper, the evaluation should more explicitly demonstrate that the inferred belief distributions correspond well to actual human beliefs rather than relying primarily on improvements in downstream collaborative performance.